# Genetic Diversity and Population Structure of *Spirobolus bungii* as Revealed by Mitochondrial DNA Sequences

**DOI:** 10.3390/insects13080729

**Published:** 2022-08-15

**Authors:** Runfeng Xu, Jie Chen, Yu Pan, Jiachen Wang, Lu Chen, Honghua Ruan, Yongbo Wu, Hanmei Xu, Guobing Wang, Hongyi Liu

**Affiliations:** 1The Co-Innovation Center for Sustainable Forestry in Southern China, College of Biology and the Environment, Nanjing Forestry University, Nanjing 210037, China; 2Key Laboratory for Ecology and Pollution Control of Coastal Wetlands (Environmental Protection, Department of Jiangsu), School of Environmental Science and Engineering, Yancheng Institute of Technology, Yancheng 224007, China

**Keywords:** genetic diversity, geographical barrier, Yangtze river, soil macrofauna, *Spirobolus bungii*

## Abstract

**Simple Summary:**

*Spirobolus bungii* plays a key role in soil ecolsystems, but there are few studies on this species, especially regarding genetics. In this study, 166 *S. bungii* individuals were collected from two cities in China (Nanjing and Tianjin), and *COX2* and *Cytb* gene sequences of mitochondrial DNA and 18S rRNA gene fragments were sequenced. We conducted a population genetic analysis based on mtDNA sequences, revealing the population genetic diversity and genetic structure of *S.*
*bungii* in the two cities, and confirming the hindering effect of geographical barriers on the gene flow of the populations.

**Abstract:**

Soil macrofauna, such as *Spirobolus bungii*, are an important component of ecosystems. However, systematic studies of the genetic diversity, population genetic structure, and the potential factors affecting the genetic differentiation of *S. bungii* are lacking. We performed a population genetic study of 166 individuals from the mountains to the south of the Yangtze River, north of the Yangtze River in Nanjing city, and near Tianjin city, in order to investigate the correlations between geographical distance and genetic diversity. A total of 1182 bp of *COX2* and *Cytb* gene sequences of mitochondrial DNA, and 700 bp of the 18S rRNA gene sequence were analyzed. There were two haplotypes and one variable site in the 18S rRNA gene, and 28 haplotypes and 78 variable sites in the *COX2* and *Cytb* genes. In this study, the 18S rRNA gene was used for species identification, and mtDNA (concatenated sequences with *Cytb* and *COX2*) was used for population genetic analysis. Structure cluster analysis indicated that the genetic structures of the different populations of *S. bungii* tended to be consistent at small geographical scales. Phylogenetic trees revealed that the haplotypes were clearly divided into three branches: the area south of the Yangtze River, the area to the north of the Yangtze River in Nanjing, and the area in Tianjin. Large geographical barriers and long geographical distance significantly blocked gene flow between populations of *S. bungii*. Our results provide a basic theoretical basis for subsequent studies of millipede taxonomy and population genetic evolution.

## 1. Introduction

Soil macrofauna is an important part of biodiversity and include diverse macroinvertebrates living in the soil. They play a key role in regulating the physical, chemical, and microbiological properties of soils [1]. Soil macrofauna contribute to forest ecosystem sustainability by increasing organic matter dynamics and altering soil physical properties [2]. Due to these substantial impacts, soil macrofauna are described as soil engineers. Most studies on soil macrofauna have mainly focused on ecological functions, with only a few focusing on biological characteristics [3,4], such as taxonomy and genetics.

In addition to annelids, platyhelminths, mollusks, and arthropods are important soil macrofauna. However, most genetic studies on soil macrofauna have been restricted to earthworms of the annelids. Furthermore, mitochondrial DNA is the most common molecular marker used in these studies [5,6]. In a previous study of soil macrofauna, mitochondrial DNA, nuclear DNA, and morphological techniques were considered as the methods of phylogenetic analysis [7,8,9,10].

As common soil macro-invertebrates, millipedes of the arthropods are one of the oldest terrestrial organisms and are widely distributed, with the earliest millipede fossils dating back to the late Silurian [11]. Millipedes contribute to soil processes, maintaining the soil fertility of mountain forests [12,13,14]. Millipedes are soil detritivores with a poor dispersal ability and are among the most important plant litter handlers in temperate forests [15]. The fragmented litter processed by millipedes is more easily degraded by other decomposers. Millipedes are used to monitor soil pollution owing to their important ecological functions [16,17]. In previous research, Jackson et al. used six genes (four mitochondrial and two nuclear) to present a phylogeny of the millipede family Xystodesmidae [9]. Piyatida et al. used two partial mitochondrial gene fragments (*COI* and 16S rRNA) and morphology to define four new *Apeuthes* species from Southeast Asia [10]. *Spirobolus bungii* is a common millipede species in the mountain forests of China. In this study, mitochondrial DNA was used as a marker to study the phylogeny and genetic diversity of *S. bungii*, to reveal the characteristics of the genetic diversity of millipedes. The 18S rRNA gene was used for species identification, to ensure that all the individuals studied were *S. bungii*.

This experimental study aimed to determine the population differentiation of *S. bungii* in eastern and northern China and to reveal the genetic structure and diversity of millipedes and their influencing factors. To study the genetic diversity of *S. bungii*, we needed to sample from two cities. First, considering the possibility of mass sampling, we selected Nanjing, East China. Second, *S. bungii* was found in the Yan Mountains of China [18]. Pan Mountain (Tianjin, China) is an offset of the Yan Mountains [19]; therefore, we chose Tianjin in North China as the second city for our study. Nanjing and Tianjin are developed cities with distinct differences [20,21]. Nanjing is located in a hilly area with green spaces of different sizes within the city [22,23]. The Yangtze River has run through the urban area of Nanjing since ancient times. Tianjin’s urban area is mostly on a plain, with only Pan Mountain at 864.4 m above sea level, located in the north of the city, with no main river. As a large river, the Yellow River runs between the two cities. Therefore we hypothesized that the seven populations of *S. bungii* distributed in different regions could be divided into three groups: the southern bank of the Yangtze River (Group A), the northern bank of the Yangtze River (Group B), and in Tianjin (Group C).

## 2. Materials and Methods

### 2.1. Sample Collection

A total of 166 *S. bungii* individuals were collected from seven sites: Zijin Mountain (32°05′N; 118°86′ E, ZJ, n = 25), Qixia Mountain (32°16′ N; 118°97′ E, QX, n = 25), Tang Mountain (32°07′ N; 119°03′ E, TS, n = 25), Fang Mountain (31°90′ N; 118°88′ E, FS, n = 25), Lao Mountain (32°10′ N; 118°60′ E, LS, n = 24), Longwang Mountain (32°19′ N; 118°71′ E LW, n = 22) in Nanjing, and Pan Mountain (40°05′ N; 117°15′ E, PS, n = 20) in Tianjin (Figure 1, Table 1). At each site, three soil samples were collected, along with individual organisms, between April and June 2021. After collection, all the biological samples were identified as *S. bungii* through identification of their physiological characteristics [24]. A portion of leg tissue was extracted from each animal, after which all individuals were released, and the collected tissue samples were stored at −80 °C until follow-up.

### 2.2. DNA Extraction and MtDNA Amplification and Sequencing

The genome DNA were extracted from the tissue samples using DNAiso Reagent (TAKARA, Beijing, China). According to the nuclear and mitochondrial sequences of *S.*
*bungii* species (GenBank accessions: AH001769.2 and MT767838.1), a set of PCR primers were separately designed for 18S rRNA, *COX2*, and *Cytb*: Spbu-18S-F (5′-GGA GAG CAA GAA TTA AGA-3′) and Spbu-18S-R (5′-TCG GTC AAT TCA CTC TAA-3′), Spbu-COX2-F (5′-TTA GTT AGA TGC TGT CAC-3′) and Spbu-COX2-R (5′-CAC AAA CAC TTC TCT TCC-3′), and Spbu-Cytb-F (5′-AAG TAC AGG ACG GTT AGA AC-3′) and Spbu-Cytb-R (5′-CTC CAT CTA ACC TGA ACG TC-3′). Each 40-μL reaction mixture contained 2 μL of template DNA (25–50 ng/μL), 20 μL of 2× Rapid Tap Master Mix (Vazyme, Nanjing, China), 1.5 μL of each primer (10 μM), and 15 μL H_2_O. The PCR procedure consisted of pre-denaturation at 95 °C for 5 min, denaturation at 95 °C for 30 s, annealing at 50–55 °C for 30 s, extension at 72 °C for 40 s, 35 cycles, and a final extension at 72 °C for 5 min. Following agarose gel electrophoresis, the PCR products were sent to the Tsingke Biotechnology Co., Ltd. (Nanjing, China) for sequencing in one direction.

### 2.3. Soil Detection

The soil pH and electrical conductivity (EC) were measured in a 1:5 soil:water solution using an electronic pH meter (pHs-25, Shanghai Leici) and a conductivity meter, respectively. The soil moisture content was measured after oven-drying at 105 °C for 24 h. The soil organic matter (SOM) was determined using the Walkley–Black potassium dichromate oxidation method [25]. The total nitrogen (TN) and available phosphorus (AP) (extracted with 0.5 mol L^−1^ NaHCO3) were analyzed using the standard Kjeldahl and Olsen methods, respectively. The available potassium (AK) extracted with 1 mol L^−1^ NH4OAc was determined using an atomic absorption spectrophotometer (TAS-990 AFG, Beijing Persee, China). Nitrate was extracted using 1 mol L^−1^ KCl via ultraviolet spectrophotometry (TU-1901, Beijing Persee, China) [26].

### 2.4. Sequence Analysis

The *COX2* and *Cytb* genes of each individual millipede were concatenated using Editseq and Seqman. DnaSP5.0 [27] was utilized to determine the indices of genetic diversity, such as haplotype diversity (*Hd*), nucleotide diversity (*pi*), and the mean number of pairwise differences (*K*). The pairwise fixation index (*F_ST_*), gene flow (haploid number of migrant, *N_M_*) values, and analyses of molecular variance (AMOVA) using Arlequin version 3.5 [28] were employed to quantify the genetic structure of the *S. bungii* population, which was then analyzed using Structure 2.3.4 software [29]. The set population was K 2–8, each K value was repeated three times, and the lengths of the burn-in periods and MCMC repeats were 200,000 and 200,000, respectively.

The results were uploaded to Structure Harvester [30], to obtain the best K value. To investigate the genetic evolution of different populations, based on the haplotypes and individuals of the population, maximum likelihood (ML) and Bayesian inference (BI) trees were constructed using PhyloSuite v1.2.2 [31]. ML analyses were performed using the TPM2u + F + G4 model in the IQ-tree. The BI analyses were performed using the HKY + F + G4 model in MrBayes 3.2.6, and run for 10 million generations, with tree sampling every 1000 generations and a burn-in of 25% trees. The phylogenetic relationships between the mitochondrial DNA (mtDNA) haplotypes of *S. bungii* were estimated from an unrooted statistical parsimony network using Popart v1.7 [32]. TBtools software was used to create a soil heat map [33].

## 3. Results

### 3.1. Genetic Diversity of S. bungii

A total of 1182 bp of *COX2* (534 bp) and *Cytb* (648 bp) gene sequences of mtDNA and 700 bp of the 18S rRNA gene sequence from 166 individuals were analyzed (GenBank accessions: OL449413–OL449447). Gene sequence analyses revealed that there were two haplotypes and one variable site in the 18S rRNA gene, and 28 haplotypes and 78 variable sites in the *COX2* and *Cytb* genes of the mtDNA (Table 1). As molecular identification may be more accurate than morphological identification, we conducted molecular identification of the 18S rRNA gene, and the results verified that all collected samples were *S. bungii*.

Although the 18S rRNA gene has often been used for phylogenetic analysis among different species [34,35,36], it has not been commonly used for individual intraspecies analysis [37]. The 18S rRNA gene sequence of all collected biological samples was relatively conservative; thus, it was not used as the basis for phylogenetic analysis in our experiment. Except for the PS population (*COX2*: pi = 0.00094, S = 5, Hd = 0.368; *Cytb*: Pi = 0.00077, S = 6, Hd = 0.447), all population data indicated that *Cytb* had a higher genetic diversity than *COX2*. In the concatenated sequence, the number of haplotypes ranged from two to nine per sampling site, with ZJ having the fewest (n = 2) and PS having the most (n =9); 12 haplotypes were found in the southern bank of the Yangtze River, seven haplotypes were found in the northern bank of the Yangtze River, and nine haplotypes were found in Tianjin, and the haplotypes in each group were unique haplotypes. Haplotype diversity (Hd) ranged from 0.153 (ZJ) to 0.653 (PS), and the nucleotide diversity (pi) ranged from 0.00013 (ZJ) to 0.0085 (PS).

### 3.2. Population Genetic Structure of S. bungii

Gene flow occurs because of contact between different populations of a species. We studied the gene flow between each population, to explore the genetic diversity and population structure of *S. bungii* and its influencing factors (Figure 2, Table 2). The range of pairwise F_ST_ values between populations varied from 0.02778 (TS and ZJ populations) to 0.99368 (ZJ and LS populations) for the mtDNA, *Cytb*, and *COX2* genes. The results showed that on the south bank of the Yangtze River, the level of gene flow among populations was relatively high (1.04729–8.74928), and genetic differentiation was not obvious (0.02778–0.19271).

The structure analyses revealed the population genetic structure of *S. bungii*. The results showed that K = 4 (the maximum value of delta K) was the most likely number for the *S. bungii* genetic clusters (Figure 3a). However, according to the actual sampling, K = 3 may have been the most likely number of *S. bungii* genetic clusters (Figure 3b). As shown in Figure 3a, the red gene cluster was almost entirely composed of samples from the ZJ, QX, FS, and TS populations; the mazarine blue gene cluster was primarily from individuals in the LS, LW, and PS populations, and the wathet blue gene cluster was mainly from individuals in the PS populations, with very few individuals in the other populations. Structure analyses demonstrated the consistency of the population genetic structure of *S. bungii* within a small geographical scale.

Hierarchical AMOVA from the geographical regions (northern and southern banks of the Yangtze River) indicated that most of the genetic variation was between groups (98.74%), whereas little genetic variation was within populations (1.02%) or among populations within groups (0.24%) (Table 3). These results imply that the Yangtze River is a significant geographical barrier, leading to differences in the genetic diversity and population structure of *S. bungii*. However, as a control with populations within a small geographical range, AMOVA from the geographical regions (Nanjing and Tianjin) identified only 32.14% of the variation as being present between groups, whereas 66.40% was among populations within groups and 1.46% was within populations (Table 3).

### 3.3. Phylogenetic Analyses and the TCS Haplotype Network

The ML tree (Figure 4a) showed that the haplotypes were clearly divided into three branches. Based on these results, three groups were considered to exist among the experimental individuals. The uppermost branch was the group of populations distributed on the southern bank of the Yangtze River (Group A), the middle branch was the group of populations located on the bank of the Yangtze River (Group B), and the bottom branch was the group of populations spread across Tianjin (Group C). The BI tree (Figure 4b) showed that Group B was more closely related to Group C than to Group A. Thus, the division of *S. bungii* populations into three groups according to their geographical distribution was supported by the results of the phylogenetic tree.

The TCS haplotype network (Figure 4c) showed that the common haplotypes in groups A, B, and C were Hap1, Hap5, and Hap13, respectively. Hap1 was distributed across four populations (ZJ, QX, TS, and FS) in Group A, Hap4 in two populations (QX and FS) in Group A, and Hap5 in two populations (LS and LW) in Group B. We observed two haplotypes in ZJ (Hap1 and Hap28), six haplotypes in QX (Hap1, Hap4, Hap21, Hap22, Hap23, and Hap24), four haplotypes in FS (Hap1, Hap2, Hap3, and Hap4) and TS (Hap1, Hap25, Hap26, and Hap27), five haplotypes in LS (Hap5, Hap6, Hap7, Hap8, and Hap9), three haplotypes in LW (Hap5, Hap10, and Hap11), and nine haplotypes in PS (Hap12, Hap13, Hap14, Hap15, Hap16, Hap17, Hap18, Hap19, and Hap20).

A dominant role of intrapopulation variations in intragroup variations was found in the analyses of these two geographical regions. This result indicated that two populations within the same geographical region (*S. bungii* populations distributed along the Yangtze River in the Nanjing areas) were more distant than those distributed in the Tianjin area. The influence of the Yangtze River on the genetics of *S. bungii* was more profound than that of the Yellow River.

### 3.4. Relationship between Environmental Factors and Haplotype Distribution

The physical and chemical properties of the soils in the different mountains of Nanjing differed to some extent (Figure 5). The relationships between different haplotypes and the corresponding soil physical and chemical properties were reflected through color gradients and similarities in the soil heat map (Figure 6). Hap1 was the most common haplotype in Group A and was distributed across every population in this group (ZJ, QX, FS, and TS). The corresponding soil physical and chemical properties represented the average values for this group. Hap28 was a unique haplotype of Ziijin Mountain (ZJ), and the corresponding soil samples had high levels of soil moisture, whereas the soil pH was below the average for Group A. Hap21, Hap22, Hap23, and Hap24 were unique haplotypes of Qixia Mountain (QX), where the available soil potassium was high. Hap2 and Hap3 were distributed across Fang Mountain (FS), and most of the physical and chemical soil properties, except for pH, were lower than the average for this group. Hap25, Hap26, and Hap27 were the normal haplotypes of Tang Mountain (TS), where the available soil phosphorus, potassium, and soil moisture showed extremely low levels.

Hap5 was the most common haplotype in Group B and was widely distributed across both populations (LS and LW). The corresponding physical and chemical soil properties represented the average value for this group. Hap6, Hap7, Hap8, and Hap9 were distributed only on Lao Mountain (LS), where the levels of organic soil carbon and soil moisture were lower than the average for Group B. Hap10 and Hap11 were distributed only on Longwang Mountain (LW), where the levels of organic soil carbon and soil moisture were above average and EC was at a significantly lower level. Hap12–20 were all found in Group C (PS), and the results showed that the physical and chemical soil properties on the Pan Mountain of Tianjin (PS) were different from those of other sample areas in Nanjing.

## 4. Discussion

### 4.1. Differences in Genetic Structure between Soil Macrofauna

Dispersal is an important ecological process, which is widely believed to play a role in determining the community structure of species, together with local factors [38,39]. There are two forms of dispersal in animals: namely, active dispersal and passive dispersal. Active dispersal has strict requirements for dispersers [40]. Individual size and the motor organs are generally thought to be related to dispersal ability [41]. Within a certain geographical range, the dispersal ability of a species affects its genetic structure and the gene flow of the population [42]. The effect of dispersal ability on genetic structure has been demonstrated in some soil animals. The limited dispersal ability of soil annelids (such as earthworms) hinders the gene exchange between their populations. Their population genetic diversity is positively correlated with habitat size [7,40,43]. Some soil insects (such as beetles and ants) are excellent at dispersal. Their wings allow them to travel further for genetic exchange with other populations [44,45].

Millipedes, as soil macrofauna, are large and have numerous pairs of legs. In the present study, we observed higher levels of gene flow (1.04729–8.74928; Table 2) on the southern bank of the Yangtze River (Group A). According to the results, *S. bungii* shows lower levels of population genetic differentiation than some soil animals proven to have limited dispersal ability (such as earthworms) [46]. Therefore, we speculate that the dispersal ability of *S. bungii* in Nanjing is relatively strong and that the habitat patches connecting habitats serve as bridges between populations, making gene exchange possible (Figure 2). On an evolutionary time scale, exchanges between populations reduce the level of population genetic differentiation.

We believe that *S. bungii* in Nanjing has a certain dispersal ability, which is, however, only one of the factors influencing the genetic diversity and structure of the population in the area. Local factors (natural and artificial habitat spatial heterogeneity) may also play a role [41]. Our current understanding of the relationship between the dispersal ability of soil animals, population differentiation, and genetic diversity is rather limited. Therefore, future research should focus on the relationship between the population genetic structure and diversification factors.

### 4.2. Gene Flow between S. bungii Populations Blocked by Geographical Barriers

*S. bungii* population analyses revealed a significant relationship between phylogeny and geography, demonstrating the presence of a phylogeographic structure. *S. bungii* exhibited a high level of genetic differentiation, with many unique haplotypes and without shared haplotypes, which showed that the *S. bungii* populations were differentiated from each other. The AMOVA results (Table 2) also corresponded to the results of the phylogenetic tree (ML and BI trees) (Figure 4a,b); namely, groups A and B were geographically close but genetically distant, and groups B and C were geographically far apart but genetically close. We speculate that the reason for this might be related to the geological timeframes of the formation of the Yangtze and Yellow rivers.

Studies on the basalt age in the Nanjing area have shown that the Yangtze river formed in the Miocene (~23 million years ago) [47,48,49]. It has been generally acknowledged that the Yellow river formed during the Pleistocene epoch (~1.15 million years ago) [50,51]. The results of the millipede phylogenetic analyses appear to be explained by the geological timeframes of the formation of the Yangtze and Yellow rivers. Recent studies have indicated that large geographical barriers have an obvious blocking effect on genetic diversity and gene exchange within populations [52,53,54]. At present, the isolation of soil fauna in the Yangtze and Yellow rivers has not been extensively described, but some studies have shown that the Yangtze and Yellow rivers block the gene exchange of some terrestrial animals [55,56].

Therefore, we considered that the obvious differences in the haplotypes of *S. bungii* might be intimately and directly related to the formation of the Yangtze and Yellow rivers. Their formation could have carved large populations into smaller ones, leading to neutral evolutionary processes such as genetic drift. Prior to the Miocene, there may have been no obvious genetic differentiation in the eastern coastal areas of China. Following the formation of the Yangtze River, the populations to the north (the predecessor species of Groups B and C) could not contact the populations to the south (the predecessor species of Group A). Over thousands of years of evolution, the genetic distance between the two populations increased, and genetic differentiation may have become evident. The formation of the Yellow River in the early Pleistocene might have hindered the gene exchange of *S. bungii* once again, and genetic differentiation may have appeared between the populations distributed to the north of the Yangtze River and in the Tianjin region. Due to the later divergence time, the genetic distance between Groups B and C was smaller than that between Groups A and B.

### 4.3. Environmental Factors on the Genetic Diversity of S. bungii

In the present study, we investigated the relationships between the environmental factors represented by the basic physical and chemical properties of the soil and the genetic diversity of *S. bungii*. Soil is the main living space that supports the activities of soil-dwelling animals [57]. Biological soil invertebrate communities influence the litter decomposition rate [58,59], and environmental factors lead to the differentiation of soil animal communities [60,61]. Thus, it is important to study the relationships between soil-residing animals, represented by *S. bungii* and the soil environment.

We speculated that several physical and chemical soil properties may influence the *S. bungii* genotypes. The results (Figure 6) revealed some heterogeneity in the soils of the different mountain locations. Taking the samples from Group A as an example, there were clear differences in the soil properties of the four mountains. The soil available phosphorus content of Tang Mountain was significantly lower than the average level, whereas the total nitrogen content of Qixia Mountain was lower, and the soil moisture content of Zijin Mountain was significantly higher. However, the distribution of haplotypes was relatively concentrated, and a universally common haplotype was present in each population.

Although there was no evident correlation between haplotype distribution and soil properties, habitat conditions had an undeniable influence on the genetic diversity of *S. bungii*. Mitochondria have the advantages of a simple structure, low molecular weight, and maternal inheritance [62]; therefore, we selected gene fragments in the mitochondrial DNA of *S. bungii* to study their genetic diversity. Our results suggested that the relationship between soil properties and millipede mitogenomes is not clear; however, the effects of other habitat conditions on genetic diversity cannot be neglected. Therefore, in subsequent experiments, we aim to use further methods, such as simple sequence repeats, and more genes, such as nuclear genes, to assess the potential impacts of habitat conditions on the genetic diversity of *S. bungii*.

## 5. Conclusions

This study focused on the genetic diversity and population structure of *S. bungii* in Nanjing and Tianjin. The results revealed that the gene flow between populations of *S. bungii* increased the consistency of the genetic structure. However, large geographical barriers, such as between the Yangtze and Yellow rivers, significantly blocked the gene flow between *S. bungii* populations, leading to high levels of genetic differentiation. Moreover, the correlation between the soil physical and chemical properties and haplotype distribution of *S. bungii* is not obvious. Our study fills research gaps in the genetic diversity of *S. bungii* at different geographical scales and provides a theoretical basis for subsequent studies of millipede phylogenetic analyses, community composition, and genetic diversity.

## Figures and Tables

**Figure 1 insects-13-00729-f001:**
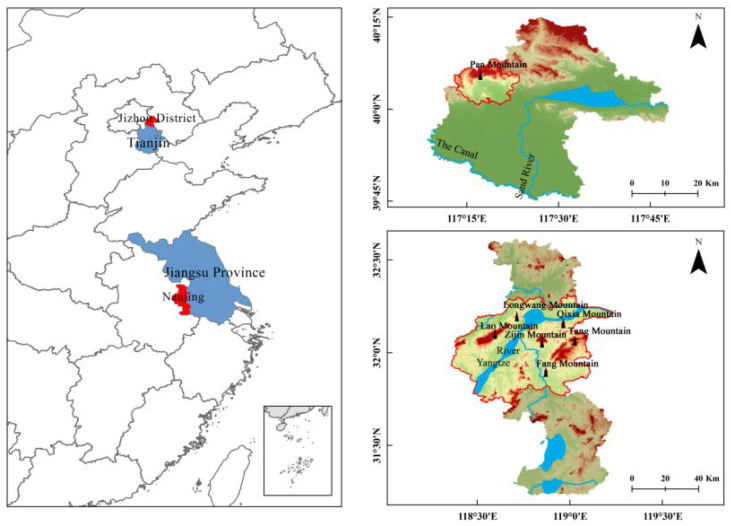
Seven sampling localities of *S. bungii* used in this study.

**Figure 2 insects-13-00729-f002:**
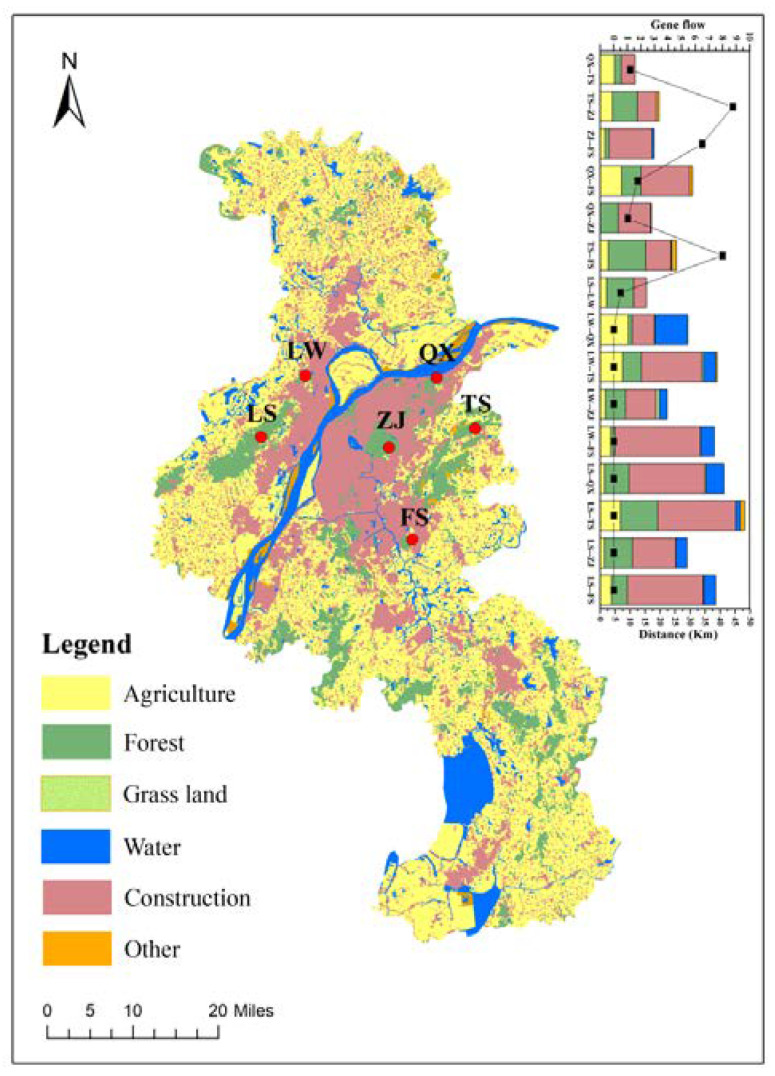
Land use in the Nanjing area and its relationship to the gene flow of *S. bungii* populations.

**Figure 3 insects-13-00729-f003:**
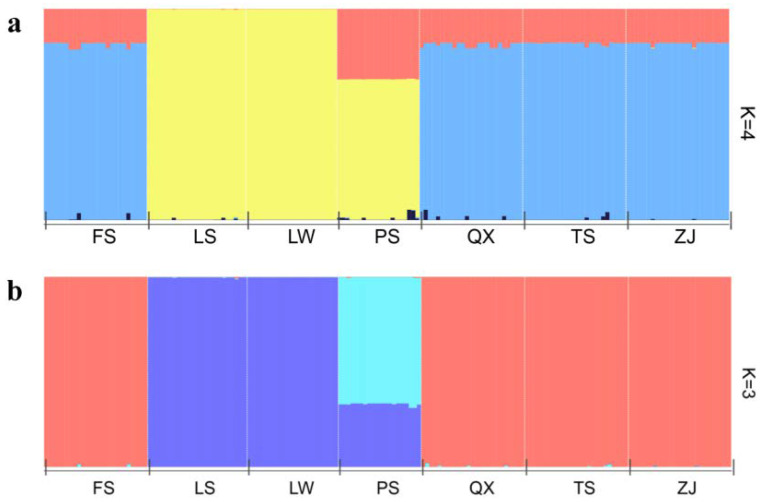
(**a**). STRUCTURE cluster analysis of seven populations of *S. bungii* (K = 4). (**b**). STRUCTURE cluster analysis of seven populations of *S. bungii* (K = 3).

**Figure 4 insects-13-00729-f004:**
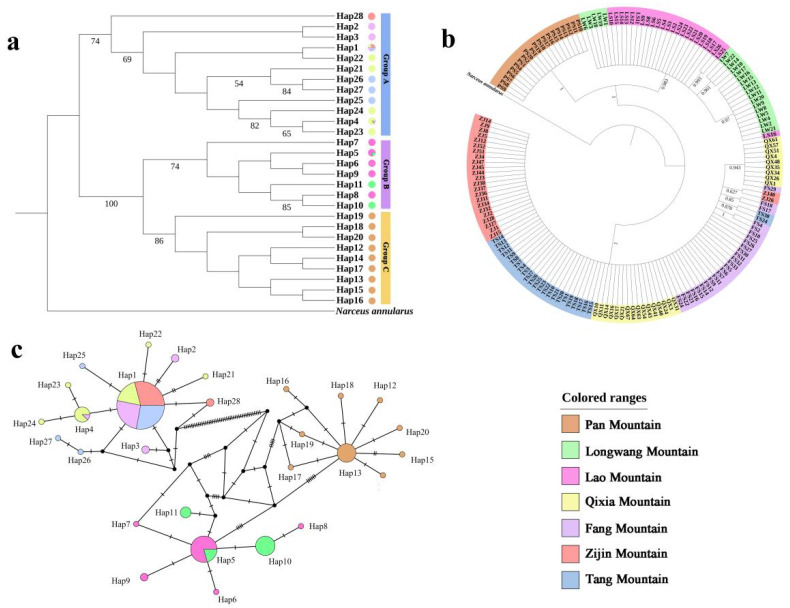
(**a**) Maximum likelihood (ML) tree based on mitochondrial *COX2* and *Cytb* tandem sequences between seven populations in *S. bungii* using 28 haplotypes. The numbers on nodes represent the supporting values for ML analysis. Supporting values on nodes below 50 are not displayed. (**b**) Bayesian inference (BI) tree based on mitochondrial *COX2* and *Cytb* tandem sequences between seven populations of *S. bungii* using 166 individuals. The numbers on nodes represent the supporting values for BI analysis. (**c**) TCS haplotype network based on *COX2* and *Cytb* genes.

**Figure 5 insects-13-00729-f005:**
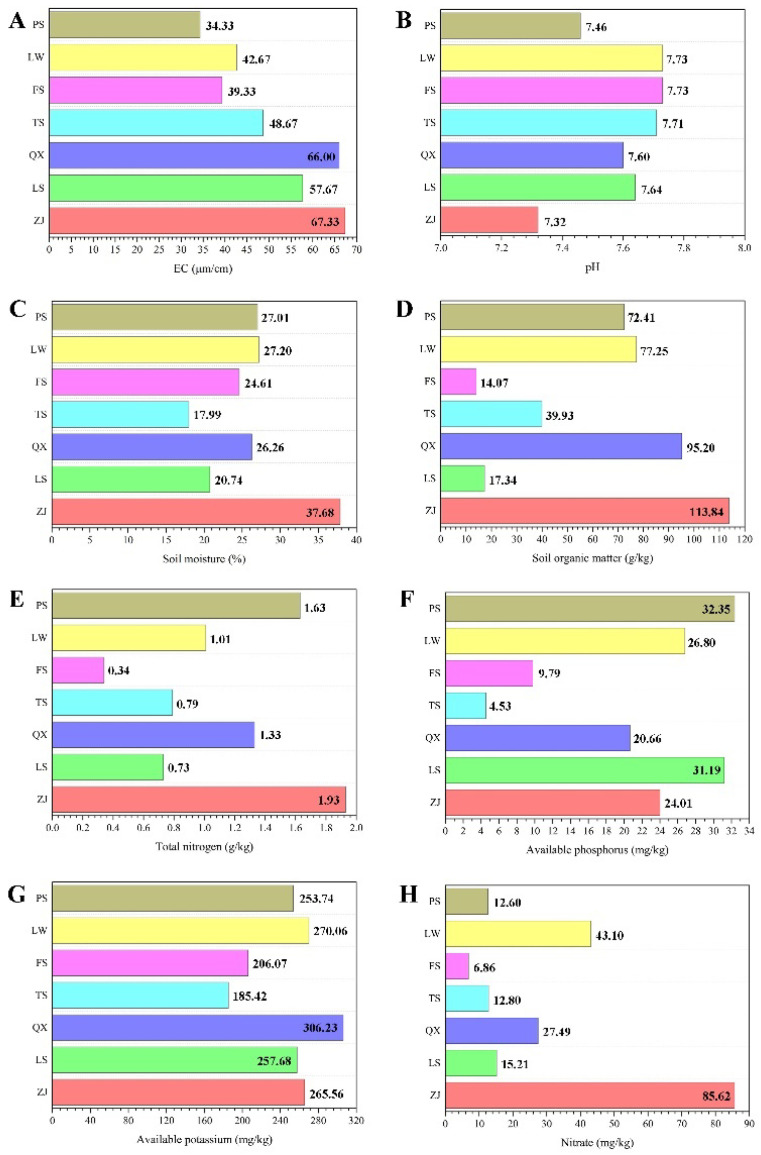
Average level of physical and chemical soil properties of the seven mountains. (**A**): The average level of EC. (**B**) The average level of pH. (**C**) The average level of Soil moisture. (**D**) The average level of Soil organic matter. (**E**) The average level of Total nitorgan. (**F**) The average level of Available phosphorus. (**G**) The average level of Available potassium. (**H**) The average level of Nitrate.

**Figure 6 insects-13-00729-f006:**
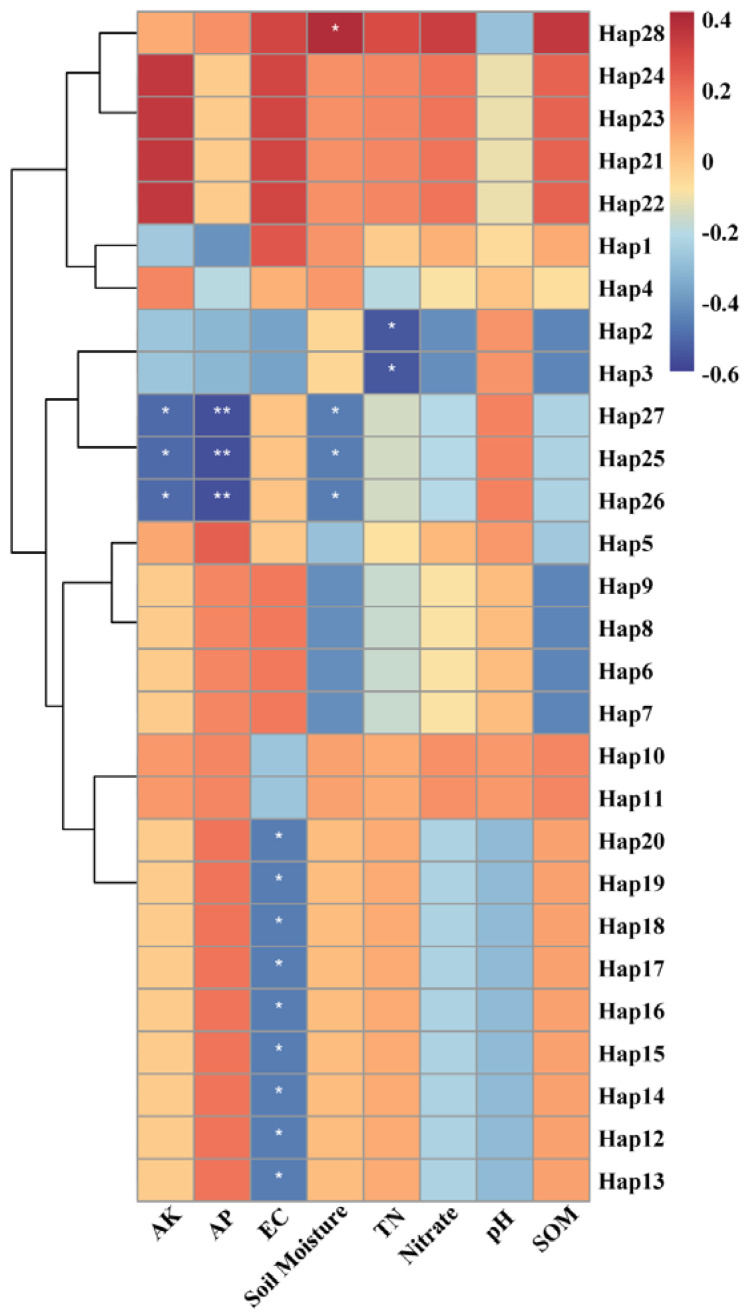
Soil heat map based on 28 *S. bungii* haplotypes and physical and chemical soil properties of seven plots. * Values for available potassium (AK), available phosphorus (AP), electrical conductivity (EC), soil moisture, total nitrogen (TN), nitrate, soil pH and soil organic matter (SOM) are given for each haplotype. Note: * <0.4 or >−0.4 for indices of physical and chemical soil properties, ** >−0.6 for indices of physical and chemical soil properties.

**Table 1 insects-13-00729-t001:** Haplotype and nucleotide diversity for mitochondrial *Cytb*, *COX2*, and nuclear 18S rRNA data of *S. bungii*.

Locality	N	*COX2* Sequence					*Cytb* Sequence					Concatenated Sequence					18S rRNA Sequence				
		*S*	*H*	*H_d_*	*pi*	*K*	*S*	*H*	*H_d_*	*pi*	*K*	*S*	*H*	*H_d_*	*pi*	*K*	*S*	*H*	*H_d_*	*pi*	*K*
South bank of the Yangtze river																					
ZiJin Mountain (ZJ)	25	-	1	-	-	-	1	2	0.153	0.00024	0.153	1	2	0.153	0.00013	0.153	1	2	0.080	0.00011	0.080
Tang Mountain (TS)	25	1	2	0.080	0.00015	0.080	2	3	0.157	0.00036	0.233	3	4	0.230	0.00027	0.313	1	2	0.153	0.00022	0.153
QiXia Mountain (QX)	25	4	4	0.230	0.00060	0.320	2	3	0.507	0.00086	0.560	6	6	0.627	0.00074	0.880	-	1	-	-	-
Fang Mountain (FS)	25	1	2	0.153	0.00029	0.153	3	4	0.360	0.00060	0.387	4	4	0.360	0.00046	0.540	1	2	0.380	0.00054	0.380
Total of south	100	6	6	0.117	0.00026	0.140	7	8	0.321	0.00059	0.380	13	12	0.371	0.00044	0.520	1	2	0.165	0.00024	0.165
North bank of the Yangtze river																					
Lao Mountain (LS)	24	2	3	0.163	0.00031	0.167	3	3	0.236	0.00050	0.326	5	5	0.377	0.00042	0.493	-	1	-	-	-
LongWang Mountain (LW)	22	-	1	-	-	-	2	3	0.593	0.00126	0.818	2	3	0.593	0.00069	0.818	-	1	-	-	-
Total of north	46	2	3	0.086	0.00016	0.087	4	6	0.604	0.00112	0.724	6	7	0.651	0.00069	0.811	-	1	-	-	-
Population in Tianjin																					
Pan Mountain (PS)	20	5	5	0.368	0.00094	0.500	5	6	0.447	0.00077	0.500	10	9	0.653	0.00085	1.000	-	1	-	-	-
Total	166	33	14	0.603	0.01933	10.324	45	19	0.717	0.02437	15.793	78	28	0.742	0.02210	26.117	1	2	0.498	0.00071	0.498

Valves for number (N), Valves for number of variable sites (*S*), number of haplotypes (*H*), haplotype diversity (*Hd*), nucleotide diversity (*pi*), and the mean number of pairwise differences (*K*) are given for each population.

**Table 2 insects-13-00729-t002:** Pairwise gene flow (above diagonal) and *F_ST_* value (below diagonal) of the seven populations of *S. bungii*.

Population	ZJ	QX	FS	TS	LS	LW	PS
ZJ	-	1.04729	6.49946	8.74928	0.00159	0.00238	0.00274
QX	0.19271	-	1.75401	1.20943	0.00336	0.00414	0.00445
FS	0.03704	0.12475	-	8.00083	0.0025	0.00332	0.00366
TS	0.02778	0.17130	0.03030	-	0.00198	0.00277	0.00312
LS	0.99368	0.98673	0.98995	0.99214	-	0.51527	0.01865
LW	0.99057	0.98372	0.98689	0.98905	0.32668	-	0.02193
PS	0.98914	0.98251	0.98558	0.98768	0.93057	0.91935	-

**Table 3 insects-13-00729-t003:** Analysis of molecular variance (AMOVA) for *S. bungii* populations based on mtDNA.

Source of Variation	d.f.	Sum of Squares	Variance Components	Percentage of Variation
**Scenario I:** the populations in the southern bank of the Yangtze River (Groups A) and the northern bank of the Yangtze River (Group B) of *S. bungii*.				
Among Groups	1	1603.880	25.42563 Va	98.74
Among populations Within groups	4	7.072	0.06143 Vb	0.24
Within populations	141	36.898	0.26355 Vc	1.02
Total	146	1647.849	25.75062	
**Scenario II:** the populations in southern bank of the Yangtze River + the northern bank of the Yangtze River (Groups A + B) and Tianjin (Group C) of *S. bungii*.				
Among Groups	1	497.325	6.40580 Va	32.14
Among populations Within groups	5	1610.952	13.23417 Vb	66.40
Within populations	160	46.398	0.29181 Vc	1.46
Total	166	2154.675	19.93178	

## Data Availability

Data available in a publicly accessible repository. The data presented in this study are openly available in NCBI at accession number OL449413–OL449447.

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
