# Peer review of "Genetic Diversity and Population Structure of Spirobolus bungii as Revealed by Mitochondrial DNA Sequences"

_insects, 2022, doi:10.3390/insects13080729_

Round 1

Reviewer 1 Report

The followings are my major or minor concerns, which should be addressed by the authors.

Title

The title is: “Genetic Diversity and Population Structure of Spirobolus bungii as Revealed by Mitochondrial DNA Sequences” despite in the manuscript it is also reported the use of 18S gene sequence (that is a part of the ribosomal RNA). I wonder what is the information that the reader can gain from the title and I’m quite confused about the use of this marker in your analysis. Please clarify this aspect especially in relation to the title.

Simple Summary

Line 15Spirobolus bungii play a key role in soil ecolsystem, but it has not received enough attention”. Please clarify the meaning of “enough annention”, from what point of view?

Lines 17-18 “COX2 and Cytb gene sequences of mitochondrial DNA and 18S rRNA gene sequence were sequenced. I suggest to replace “sequence” with fragments. Sequences were sequenced is somewhat repetitive.

Line 18 “We conducted population genetic analysis based on mtDNA sequences, revealing…” Please clarify the use of the two different markers (mitochondrial and nuclear).

Abstract

Line 21 “Soil macrofauna, such as Spirobolus bungii, are an important…” Replace with “is” an important….

Line 24 “study of 166 individuals on the mountains to the south…” Replace with “from” the mountanins to…

Line 25 “and near Tianjin city to investigate the correlations between….”Replace with “in order to” investigate….

Line 28 “There were two haplotypes and one variable site in the 18S rRNA gene and 28 haplotypes and 78 variable sites in the COX2 and Cytb………” I understand that the 18S ribosomal RNA gene is involved, but sometimes it is referred to as "18S gene" and others as “18S rRNA gene”, I would recommend the authors standardize the terminology throughout.

Lines 32-33 “Large geographical barriers and long geographical significantly blocked gene flow between populations of S. bungii. “Please rephrase this sentence. The meaning is unclear.

Introduction

Line 40 “Soil macrofauna are an important..” Replace with “is” an important….

Lines 46, 47, 48 “It is difficult to study and use soil macrofauna to improve the soil environment without an understanding of their biological characteristics.” I'm quite confused about the meaning of this sentence. Please rephrase

Lines 52, 53, 54 “Without more molecular markers developed for soil macrofauna, mitochondrial DNA still provides valuable information on their phylogeny and genetic diversity [5-10]”. On the use of only two mitochondrial markers to estimates the genetic diversity, I have a couple of questions: I admit I am not very familiar with millipedes and the members of the families and genera of Spirobolida, but I have found four references on Phylogeny of the millipede order Spirobolida that combines morphological and DNA data. Particularly they used both mitochondrial (COI, 12S, tRNA-Val and 16S rRNA) and nuclear (18S and partial 28S ribosomal RNA genes) markers. The authors seem to disregard or neglect some important markers that have been used in this specific field. For example authors should not ignore the following:

·     Pitz, K. M., & Sierwald, P. (2010). Phylogeny of the millipede order Spirobolida (Arthropoda: Diplopoda: Helminthomorpha). Cladistics, 26(5), 497-525.;

·     Pimvichai, P., Enghoff, H., Panha, S., & Backeljau, T. (2020). Integrative taxonomy of the new millipede genus Coxobolellus, gen. nov. (Diplopoda: Spirobolida: Pseudospirobolellidae), with descriptions of ten new species. Invertebrate Systematics, 34(6), 591-617.;

·     Means, J. C., Hennen, D. A., Tanabe, T., & Marek, P. E. (2021). Phylogenetic systematics of the millipede family Xystodesmidae. Insect Systematics and Diversity, 5(2), 1.:

·     Pimvichai, P., Panha, S., Backeljau, T., & Giribet, G. (2022). Combining mitochondrial DNA and morphological data to delineate four new millipede species and provisional assignment to the genus Apeuthes Hoffman & Keeton (Diplopoda: Spirobolida: Pachybolidae: Trigoniulinae). Invertebrate Systematics, 36(2), 91-112.

I think the authors should still address this subject rather than simply saying " Without more molecular markers developed for soil macrofauna, mitochondrial DNA still provides valuable information on their phylogeny and genetic diversity”

Lines 55-56 “As common soil macro-invertebrates, arthropod millipedes of arthropods are one of the oldest terrestrial organisms and are widely distributed worldwide,…” arthropod millipeds of arthropods and wideley distribuite worldwide are somewhat repetitive. Please rephrase.

Lines 62-63 “However, to the best of our knowledge, only a few genetic studies have been performed on these organisms” Please add references (see the comment about references).

Lines 64,65 and 66In this study, mitochondrial DNA was used as a marker to study the phylogeny and genetic diversity of S. bungii to reveal …………” Please make clear to the reader the meaning of the use of the18S ribosomal RNA gene in this study.

Line 69 “To study the genetic diversity of S.bungii, we needed to sample from two cities.” What was the criteria for defining the study sites? I mean it seems strange that to study the genetic diversity of a whole species, a sampling campaign from only two cities is enough. Does the lack of samples from other certain regions (within the distribution area) represent any kind of bias that might influence the results? I think the authors should still better explain this subject.

Results

Figure 1. “Three sampling localities of S. bungii used in this study.” The manuscript reports that the specimens were collected from seven sites (in two cities). I found this caption confusing.

Figure 4. I suggest highlighting the three groups A, B and C in fig. 4a, e.g. using tre bars (A, B and C) on the right side of the tree. Moreover, looking at fig. 4b, I suggest keeping the orientation of clades in these figures consistent, so that the tree topology and associated color schemes are standardized - makes it easier for the reader to follow the patterns, e.g. use for Fig. 4b the same rectangular layout as in Figure 4a. Finally, I suggest adding to the caption the meaning of the numbers reported along the branches.

Discussion

Line 316 “Local factors (natural and artificial habitat spatial heterogeneity) may also play a role.” Are there any studies on this topic?

Another point is that the biogeographic scenario presented in the Discussion seem largely ad hoc, “The geological timeframes of the formation of the Yangtze and Yellow rivers were consistent with the results of the millipede phylogenetic analyses” i.e. there is no formal analysis of any kind, neither a divergence time estimation using molecular data (general arthropod molecular clock) or fossils, in order to validate the age of the three groups (A, B and C).

Finally, there are rather numerous sentences in which the English could be improved for clarity.

Reviewer 2 Report

If I had one suggestion for this study, the analysis of ZJ, QX, FS, and TS as different groups is erroneous. It is somewhat unreasonable to treat samples from these regions as a population genetics tool like AMOVA and Structure. Some corrections are needed in the following comments.

Minor comments:

Line 83-90. Sample collection information is insufficient. There is no detailed information about exact collection date, GPS, and etc. Is there any permission required to collect samples in those sites?

Line 125-126. Alignment method of sequences was not described. How did you make the concatenated sequence with 18S in Editseq and Seqman? How did you treat gap sequence without other software?

Line 168. In table 1, what is the total value in a raw of K?

Line 168 (Table 1), 214. South of the Yangtze river = southern bank of the Yangtze river? North of the Yangtze river = northern bank of the Yangtze River? Please unify the wording.

Line 177-179. How to calculate gene flow? There is no information in M&M. How did you determine the level of gene flow? Is there any reference? According to figure1 and table2, I think FS, TS, QX, and ZJ are collected from the same site not being isolated physically. This is a critical problem of sampling.

Lin 186-187. “K = 4 (the maximum value of delta K) was the most likely number for 186 S. bungii genetic clusters” You should suggest the result of Structure Harvester in supplementary material.

Line 283. S. bungii -> be italicized

Line 305-307. I think FS, TS, QX, and ZJ are almost same within a population. It is awkward that there is the gene flow between them.

Line 332-341. This part is fancy. If you could calculate the divergence date between groups A and B, it will be better to support the separation geographically by bifurcate event of the Yangtze river.

Line 353-354. This is a conclusion of this study.

-      End  -

Reviewer 3 Report

The presented paper concerns interesting aspects of studying soil macrofauna like Spirobolus bungii. However, it needs some more improvements.

- in the methods section - DNA extraction and mtDNA amplification and sequencing change the amount of added regents to the concentration that's why other researchers could repeat your experiments

- in Sequence analysis model for BI analysis is lacking, add it

- why combined data from COX2, CytB and 18S was not used for phylogenetic analysis? that way it would be more reliable

-Figure 4a and 4b- results of ML and BI analysis are hard to compare due to different ways of presentation- different phylograms. It would be better if they were presented in the same way. Why in ML analysis haplotypes were used as an input data and individuals in BI ? Supporting values on nodes below 50 should not be displayed on ML phylogram.

-In discussion, you are speculating about the genetic distance between those populations from group B/C and A/B, have you estimated it based on obtained DNA sequences or this is only divagations

Round 2

Reviewer 1 Report

The manuscript is much improved but there is one thing that should be addressed before final acceptance. Please see this minor point below.

Figure 4. I suggest keeping the orientation of clades in these figures consistent, so that the tree topology and associated color schemes are standardized - makes it easier for the reader to follow the patterns, e.g. use for Fig. 4b the same  rectangular layout as in Figure 4a. In the present form the results of ML and BI analysis are hard to compare. This was indicated in my earlier comments. Reviewer has also touched on this issue.